# Immunohistochemical Characterization of Immune System Cells in Lymphoid Organs from Roe and Fallow Deer

**DOI:** 10.3390/ani12213064

**Published:** 2022-11-07

**Authors:** Niccolò Fonti, Francesca Parisi, Francesca Millanta, Maria Irene Pacini, Marcello Periccioli, Alessandro Poli

**Affiliations:** 1Dipartimento di Scienze Veterinarie, Università di Pisa, Viale delle Piagge 2, 56124 Pisa, Italy; 2Unità Funzionale di Sanità Pubblica Veterinaria e Sicurezza Alimentare Zona Distretto Grossetana, Dipartimento di Prevenzione, Azienda USL Toscana Sud Est, Amiata Grossetana e Colline Metallifere, Viale Cimabue, 109, 58100 Grosseto, Italy

**Keywords:** immunohistochemistry, roe deer, fallow deer, immune cells, lymph node, spleen, macrophage polarization, IHC

## Abstract

**Simple Summary:**

Diseases emerging from wildlife represent a growing public health issue. Cervids share many pathogens with domestic species and humans, representing useful spontaneous models to evaluate host-pathogen balance. Histology and immunohistochemistry can help in fully understanding the pathogenesis of infection in these species, but few studies have been conducted to characterize immune cell markers. This study highlights that lymphocytes and macrophagic subsets in roe and fallow deer lymphoid tissue can be identified by a panel of commercial antibodies developed against humans. A description of the main immune cell distribution was provided. These results may support future investigations on immune cell response and pathogenesis in roe and fallow deer diseases.

**Abstract:**

Roe and Fallow deer are common wild ruminants widely distributed in Italy. Infectious diseases of these species can potentially pose health risks to domestic animals and humans. However, few studies have been conducted in which immune system cells in these species were phenotyped. The aims of this study were to determine the cross-reactivity of a wide anti-human panel of commercial antibodies on formalin-fixed and paraffin-embedded (FFPE) samples and to describe the distribution of roe and fallow deer main immune cell subsets in the lymph nodes and spleen. Twenty retromandibular lymph nodes (RLNs) and spleen samples were collected from 10 roe deer and 10 fallow deer and were tested by a panel of 12 commercial anti-human antibodies. The CD79a, CD20, CD3, Iba-1, MAC387, and AM-3K antibodies were successfully labeled cells in cervine tissue, while the Foxp3 and the CD68 did not show suitable immunostaining. This study supplies the first immunohistochemical description of immune cell subpopulations in non-pathological spleen and RLNs from roe and fallow deer and provides an easily repeatable manual IHC protocol to immunolocalize cervine B-, T-cells, and macrophages subsets in FFPE tissue samples.

## 1. Introduction

Roe deer (*Capreolus capreolus*) and Fallow deer (*Dama dama*) are common wild ruminants widely distributed in Europe. In Italy, they represent, together with the red deer (*Cervus elaphus*) and the wild boar (*Sus scrofa*), the main species of ungulates, with a consistency of over 400,000 head for roe deer and 20,000 head for fallow deer [1]. Among wild animals, ruminants play a major role in terms of multiple routes of contact with humans and the strong influence they still have on our culture [2]. In recent decades, the interest in wildlife diseases has increased, with the awareness of their public health implications and the importance of active surveillance [3,4]. Infectious diseases of wildlife can potentially pose health risks with multiple economic and health repercussions on domestic populations, as well as on human health [5].

From a One Health perspective, it is not possible to understand epidemiology, pathogenesis, and how to control most of the diseases if diagnostic and prophylactic efforts are restricted to domestic species. Wildlife is a reservoir in which both emergent or well-known pathogens can often circulate, mutate to adapt to the new host, and finally rejoin the domestic host. Examples in deer are the spread of Tuberculosis caused by *Mycobacterium bovis* [6], Johne’s disease caused by *Mycobacterium avium* subsp. *paratuberculosis* [7,8], pathogenic *Chlamydia* sspp. and Chlamydia-like organisms [9], hepatitis E virus (HEV) [10,11,12], and border disease virus (BDV) [13].

Wild ruminant diseases could also represent useful models to evaluate the complexity of the balance between host and pathogens and the molecular factors involved. A good example is the relationship between the high morbidity and mortality rates in some ungulate species naturally infested by *Sarcoptes Scabiei* and differences in immune cell distribution in pathological tissues [14,15]. According to this, the investigation of histopathological changes and relative inflammatory infiltrates can have a crucial role.

Moreover, serological and molecular investigations are often used to carry out studies on wildlife. The preservation of these biological specimens is a challenge, particularly in remote locations or resource-limited settings where the cold chain cannot be maintained [16]. In this framework, histology and immunohistochemistry (IHC) from formalin-fixed specimens can overcome some storage issues and provide complementary data to fully understand the pathogenesis of infection in these species [17].

Unfortunately, few studies have been conducted on haemal nodes and pathological tissues from roe deer [18,19,20] and fallow deer [21,22] in which the immune system cells have been phenotyped. Furthermore, no commercial antibodies specifically developed against antigens of these species are available.

The aims of this study were (I) to determine the cross-reactivity and reliability of an anti-human panel of commercial antibodies on routine formalin-fixed and paraffin-embedded samples (FFPE), and (II) to describe the distribution and immunohistochemical localization of roe and fallow deer main immune cell subsets in the lymph nodes and spleen.

## 2. Materials and Methods

Twenty retromandibular lymph nodes (RLNs) and spleen samples were collected during the 2020–2021 hunting season, in the months of February and March, from 10 roe deer (4 males and 6 females) and 10 fallow deer (6 males and 4 females). The subjects originated from the province of Grosseto, an area characterized by the abundant presence of wild ruminants [1], and were shot following the Regional Hunting Law (Regolamento di attuazione della legge regionale 12 gennaio 1994 n. 3 DPGR 48/R/2017) and immediately carried to a game-meat storing and processing establishment.

Each animal underwent macroscopic examination, and the absence of signs of infectious or parasitic disease was set as an inclusion criterion. The samples were also examined histologically, and those that showed obvious pathological changes were excluded. All deer included in the study were completely developed and were classified according to the age as 11 adults (>2 years) and 9 subadults (1–2 years) [23]. Samples from spleen and RLNs were placed into 10% neutral buffered formalin (pH 7.4) for 48–72 h, stored at room temperature and transported to Anatomic Pathology Service of Veterinary Science Department, University of Pisa. After a routinely embedding procedure in paraffin, 4 μm-thick histological sections were cut on the microtome and stained with a routine Hematoxylin-Eosin staining protocol (H-E). The 4 μm histological sections, intended for the labeled streptavidin-biotin (LSAB) method of IHC, were mounted on polarized microscope slides Menzel-Gläser Superfrost plus (Thermo Fisher Scientific, Waltham, MA, USA) and placed at 37 °C for 24 h. Heat-mediated antigen retrieval in 250 mL of sodium citrate buffer (pH 6.0) was performed for 15 min using a microwave oven.

After being mounted on Shandon Coverplate supports (Thermo Fisher Scientific, Waltham, MA, USA), the sections were incubated for 10 min with BLOXALL Blocking Solution SP-6000 (Vector Laboratories, Newark, CA, USA) and 5 min with UltraVision Protein Block solution (Thermo Fisher Scientific, USA) to avoid endogenous peroxidase activity and non-specific antigen binding, respectively.

Twelve commercial anti-human monoclonal and polyclonal primary antibodies were tested on serial sections and incubated overnight at room temperature. Table 1 lists the antibody data and dilution: some of the antibodies used were chosen based on bibliography, while others had never been tested in the species under study [18,19,21].

After incubation for 20 min with a horse secondary biotinylated antibody (Horse Anti-Mouse/Rabbit IgG (H + L), Biotinylated, R.T.U., Vector Laboratories, USA), a streptavidin-peroxidase enzyme complex (R.T.U. Horseradish Peroxidase Streptavidin, Vector Laboratories, USA) was added for 15 min.

The colorimetric reaction was developed by adding 3-1-diaminobenzydine (ImmPACT DAB Peroxidase Substrate kit, Vector Laboratories, USA) and counterstaining with hematoxylin was performed.

Serial sections of non-pathological FFPE dog lymph nodes from the Animal Cancer Registry of the University of Pisa, in which cross-reactivity was already known [24,25], were used as a positive control. As a negative control, the primary antibodies were replaced with an irrelevant and isotype-matched antibody. Sections were observed under a light microscope. The number of positive cells in the different anatomical structures was evaluated by a 5-tier semiquantitative score (-, no; +, <10%; ++, 10–50%; +++, 50–90%; ++++, >90% immunoreactive cells) for each marker [26].

## 3. Results

Following the H-E histologic examination, well-preserved tissue morphology and architecture were evident both in lymph nodes and spleen. Lymphocytic and monocytic populations were successfully immunolocalized through eight markers; antibodies known to recognize CD79a, CD20, CD3, Iba-1, MAC387, and CD163 in human and dog tissues successfully labeled cells in cervine RLN and spleen to varying degrees of intensity and the yield was similar in both species. Suitable immunostaining was not obtained for the two anti-CD79a antibodies, the anti-Foxp3 antibody, or the anti-CD68 antibody.

### 3.1. B-Cells

In H-E stained lymph node cortex sections of both roe and fallow deer, secondary lymphatic follicles were highlighted (Figure 1A,B). In the lymph nodes, CD79a + cells were distributed mainly within the lymphatic follicles at the cortical level both in roe (Figure 1C) and fallow deer (Figure 1D) with a cytoplasmic staining pattern. Scattered immunoreactive cells in the paracortical and medullary areas were also highlighted. The Exbio antibody showed a higher percentage of immunopositive cells than the Novusbio antibody in both species. Especially for the latter, a greater immunoreactivity was found in cells from the mantle area rather than the germinal center. In the spleen, the positive cells were distributed mainly in the white pulp, within the lymphatic follicles with a pattern similar to that seen in the lymph node.

CD20 + cells also showed a cytoplasmic staining pattern and a major distribution in the lymphatic follicles (both lymph node and splenic), as reported for CD79 + cells, but with less background staining. Moreover, a higher prevalence of immunopositive cells in the inner areas of the germinal centers in both roe deer (Figure 1E) and in fallow deer (Figure 1F) was seen.

### 3.2. T-Cells

In paracortical areas of H-E stained lymph node sections of both roe and fallow deer, a mantle of small lymphocytes was highlighted (Figure 2A,B). In the cortical and paracortical zone of lymph nodes of both species, a large amount of CD3 + cells (Figure 2C–E) were detected by monoclonal and polyclonal antibodies. A cytoplasmic and membranous immunohistochemical staining pattern was observed in both antibodies. The subcapsular and interfollicular areas had the greatest distribution of these cells, but single positive cells or small clusters were also found within the germinal centers of the lymphatic follicles and at the level of the medullary sinuses (Figure 2F). CD3 + cells were also found both in the white pulp and in the splenic red pulp, mainly within the periarteriolar lymphocyte sleeves (PALS). Dako polyclonal antibody had a slightly higher prevalence of immunopositive cells than Santa Cruz monoclonal antibody, especially in the paracortical zone and PALS.

### 3.3. Macrophages

In the medullary cords of H-E stained lymph node sections of both roe and fallow deer, several large mononuclear cells were detectable (Figure 3A,B). Iba-1 + cells were abundantly found in all lymphatic tissues in both roe (Figure 3C) and fallow deer (Figure 3D). The cells had a homogeneous cytoplasmic pattern and were mainly distributed in the medullary sinuses and in the splenic cords of the red pulp in the spleen. Iba-1 + cells were also present in the cortical areas within the follicles and near the subcapsular sinuses.

The immunostained cells with the anti-calprotectin (MAC387) (Figure 3E,F), and the anti-CD163 (AM-3K) (Figure 3G,H) antibodies showed an intracytoplasmic pattern and a microscopic distribution like Iba-1 + cells. In particular, CD163 + cells were more abundant in the red pulp and medulla, while no immunostained cells were seen in lymphatic follicles. Scattered granulocytes MAC387 + were also distinguishable in both species, especially in the red pulp of the spleen, inside splenic sinusoids.

The results of the study are collected in the following table (Table 2).

## 4. Discussion

This study supplies the first immunohistochemical description of immune cell subpopulation distribution in roe and fallow deer’s spleen and lymph nodes. The localization and distribution of the principal cells of the immune system of these species were similar to that reported in domestic ruminants [27,28,29] and in the few studies conducted on secondary hemolymphatic cervid organs like RAMALT and LMN of red deer [30] and haemal nodes of roe deer [20]. Similar investigations were also carried out on peripheral lymphoid tissue in many other wild mammalian species [31], including chamois [14], raccoon [26], Iberian lynx [32], Tasmanian devil [33], and tammar wallaby [34] among others. These findings offer the basic understanding of immune cell populations in healthy animals needed for detailed characterization of the immune response and immunopathological changes in infectious and non-infectious diseases affecting this wild species.

Monoclonal and polyclonal commercial antibodies directed against human CD79a, CD20, CD3, Iba-1, MAC387, and AM-3K epitopes revealed a cross-reaction with roe and fallow deer analogs both in formalin-fixed RLNs and spleen.

Despite a significant increase in numbers in Europe, diagnostic investigation in roe and fallow deer has been limited [35,36]. Currently, fixation with 10% neutral buffered formalin is still representing the most used way to prevent tissue degradation, by the formation of covalent binding and inhibiting endogenous and exogenous enzymatic processes [37]. On the other hand, these cross-linkages can lead to epitope masking. Therefore, it is not unusual to use alternative specimens, e.g., frozen, alcohol, or zinc salt fixed tissues, to achieve a better antigen exposure, especially for immune cell lineages-related markers [38]. Despite having provided a better sensitivity in immunostaining some specific antigens, these fixation methods are still far from replacing formalin fixation, on which major laboratory activities are based. So, optimizing an easy and less expensive IHC protocol intended to immunophenotyped wild ruminants’ immune system subsets and fit on routinely fixed and processed tissues could enhance studies about disease pathogenesis in these species.

This study shows that roe and fallow deer B-cells, T-cells, and macrophages can clearly be immunolocalized in FFPE sections as well. Routinely and common antigenic unmasking methods (boiling in sodium citrate) and commercial primary antibodies readily available by most of the veterinary histopathological analysis laboratories were employed.

If employed in addition to the widely used serological and molecular methods, diagnostic techniques like IHC and in situ hybridization (ISH) allow us to provide valuable epidemiologic and pathologic information [39,40,41]. Regarding epidemiologic surveys, the risk of false-positive cases following the use of highly sensitive methods such as PCR can be drastically reduced. The localization of pathological alterations associated with the presence of a pathogen thus makes it possible to distinguish the subjects affected by cases of contamination or pollution during the analytical phases, providing a complementary test [17].

From a pathological point of view, even the most modern molecular techniques such as NGS, involve the destruction and lysis of tissue samples, with no information about the localization of certain targets. Especially in chronic pathologies and those symptomatologically inapparent, understanding the localization, distribution and atypical characteristics of the host immune system and their dialogue with the etiological agent (bacteric, viral, prionic, parasitic or neoplastic) becomes essential [42]. In this framework, macrophages are key effectors of tissue homeostasis, and their polarization into classically activated (M1) and alternatively activated (M2) macrophages is increasingly recognized as an important topic in both neoplastic and inflammatory diseases [43]. M1-polarized phenotype is associated with pro-inflammatory activity, T helper 1 (Th1) immune response boosting and antineoplastic activity, while the M2-polarized phenotype is associated with tissue repair, remodeling, and enhancement of Th2 immune response [44]. In a variety of animal species, Calprotectin and CD163 are used, respectively, as markers of M1/pro-inflammatory and M2/anti-inflammatory macrophages [45].

To our knowledge, the anti-AM-3K antibody, whose ability to identify the CD163 has been demonstrated elsewhere [25], has never been employed on cervids. On the other hand, MAC387 + cells were previously described in deer [19,21]. Based on these findings, this is the first paper in which preliminary data regarding an immunohistochemical M1/M2 polarization assessment for cervine cells are provided. Additional analytical methods, such as Western blot (WB) or protein sequencing by mass spectrometry, could be used to demonstrate the detection of identical epitopes. Nonetheless, lymphoid region distribution, cell morphology, and staining patterns of the tested antibodies allow us to confirm the specificity of our findings, as reported elsewhere [14,19,26,30].

## 5. Conclusions

In conclusion, our study represents the first immunohistochemical description of roe and fallow deer secondary lymphoid organs without recognizable pathological changes and the immunophenotyping of their major immunological subpopulations. Additionally, it provides an easily repeatable IHC protocol suitable for immunolocalizing deer B-cells, T-cells, total macrophages, and M1/M2-like macrophages that could be used for other research or diagnostic purposes in deer formalin-fixed and paraffin-embedded tissue samples.

## Figures and Tables

**Figure 1 animals-12-03064-f001:**
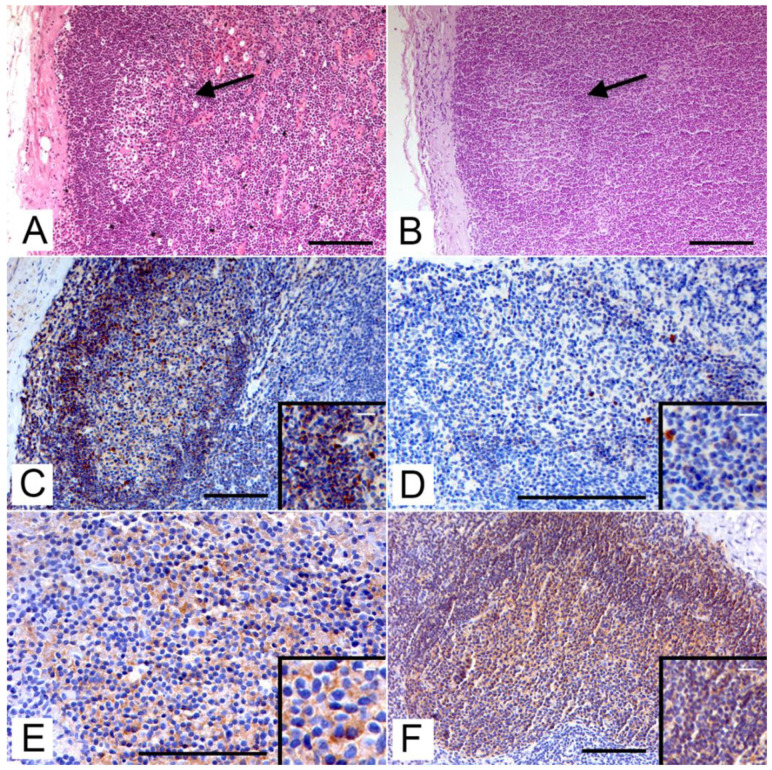
Immunolocalization of B-cells on lymph node tissue cortex with secondary lymphatic follicles by immunohistochemistry (IHC). (**A**) Roe deer. Secondary lymphoid follicle (arrow) in lymph node cortex (H-E, bar = 100 μm). (**B**) Fallow deer. Secondary lymphoid follicle (arrow) in lymph node cortex (H-E, bar = 100 μm). (**C**) Roe deer. Immunohistochemical staining with anti-CD79a (Exbio) antibody in lymphatic cells within the follicle (IHC, bar = 100 μm). Inset: CD79a-positive B lymphocytes in the mantle zone (IHC; bar = 25 μm). (**D**) Fallow deer. Slight immunohistochemical staining with anti-CD79a (Novusbio) antibody in the mantle area (IHC, bar = 100 μm). Inset: CD79a-positive B lymphocytes in the mantle zone (IHC; bar = 20 μm). (**E**) Roe deer. Immunohistochemical staining with anti-CD20 antibody (IHC, bar = 100 μm). Inset: CD20-positive B lymphocytes in the germinal center (IHC; bar = 10 μm). (**F**) Fallow deer. Immunohistochemical staining with anti-CD20 antibody (IHC, bar = 100 μm). Inset: CD20-positive B lymphocytes in the germinal center (IHC; bar = 25 μm).

**Figure 2 animals-12-03064-f002:**
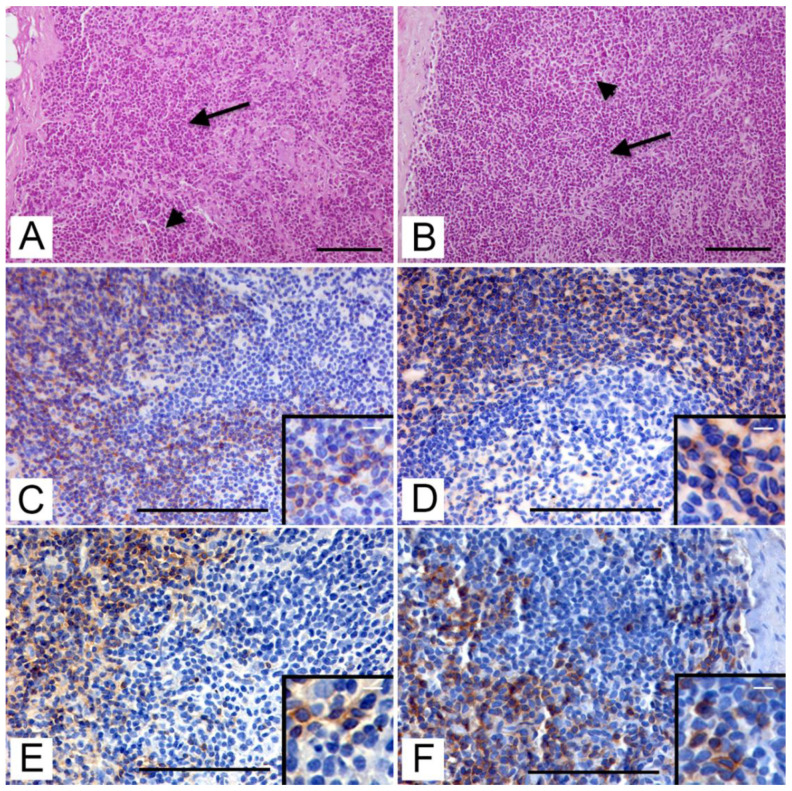
Immunolocalization of T-cells on lymph node tissue by immunohistochemistry (IHC). (**A**) Roe deer. Interfollicular area (arrow) and lymphatic follicle (arrowhead) in lymph node cortex (H-E, bar = 100 μm). (**B**) Fallow deer. Interfollicular area (arrow) and lymphatic follicle (arrowhead) in lymph node cortex (H-E, bar =100 μm). (**C**) Roe deer. Immunohistochemical staining with anti-CD3 (Santa Cruz) antibody in the interfollicular area of the cortex (IHC, bar = 100 μm). Inset: CD3-positive T lymphocytes in the interfollicular area (IHC; bar = 15 μm). (**D**) Fallow deer. Immunohistochemical staining with anti-CD3 (Dako) antibody bordering a lymphatic follicle (IHC, bar = 100 μm). Inset: CD3-positive T lymphocytes in the interfollicular area (IHC; bar = 10 μm). (**E**) Roe deer. Immunohistochemical staining with anti-CD3 (Dako) antibody bordering a lymphatic follicle (IHC, bar = 100 μm). Inset: CD3-positive T lymphocytes in the interfollicular area (IHC; bar = 10 μm). (**F**) Fallow deer. Immunohistochemical staining with anti-CD3 (Dako) antibody with scattered immunopositive cells inside the germinal center of the follicle (IHC, bar = 100 μm). Inset: scattered CD3-positive T lymphocytes in the germinal center (IHC; bar = 10 μm).

**Figure 3 animals-12-03064-f003:**
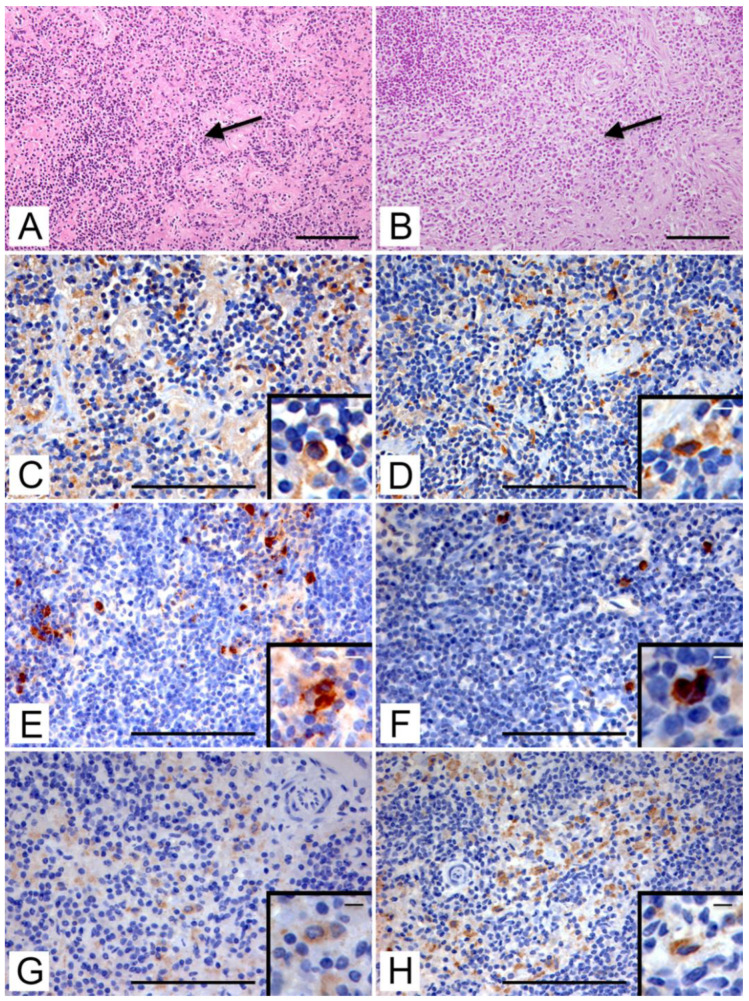
Immunolocalization of macrophages on lymph node tissue. (**A**) Roe deer. Lymph node medullary area (arrow; H-E, bar = 100 μm). (**B**) Fallow deer. Lymph node medullary area (arrow; H-E, bar = 100 μm). (**C**) Roe deer. Immunohistochemical staining with anti-Iba-1 antibody in the medulla (IHC, bar = 100 μm). Inset: Iba-1-positive macrophagic cells in the medullary area (IHC; bar = 10 μm). (**D**) Fallow deer. Immunohistochemical staining with anti-Iba-1 in the medulla (IHC, bar = 100 μm). Inset: Iba-1-positive macrophagic cells in the medullary area (IHC; bar = 10 μm). (**E**) Roe deer. Immunohistochemical staining with anti-Calprotectin (MAC387) antibody in macrophages and granulocytes in the medulla (IHC, bar = 100 μm). Inset: MAC387-positive macrophagic cells in the medullary area (IHC; bar = 15 μm). (**F**) Fallow deer. Immunohistochemical staining with anti-Calprotectin (MAC387) antibody in scattered macrophages and granulocytes in the medulla (IHC, bar = 100 μm). Inset: MAC387-positive macrophagic cells in the medullary area (IHC; bar = 10 μm). (**G**) Roe deer. Immunohistochemical staining with anti-CD163 (AM-3K) antibody in the medulla (IHC, bar = 100 μm). Inset: CD163-positive macrophagic cells in the medullary area (IHC; bar = 10 μm). (**H**) Fallow deer. Immunohistochemical staining with anti-CD163 (AM-3K) antibody in the medulla (IHC, bar = 100 μm). Inset: CD163-positive macrophagic cells in the medullary area (IHC; bar = 10 μm).

**Table 1 animals-12-03064-t001:** The 12 anti-human monoclonal and polyclonal antibodies employed and their targets, clones, species of origin, dilution, and source. CD, cluster of differentiation; Foxp3, forkhead box P3; Iba-1, ionized calcium-binding adaptor molecule 1.

Cell Marker	Cell Target	Clone	Species	Dilution	Source
B lymphocytes	CD79a	HM57	mouse mAb	1:100	Dako
	CD79a	HM57	mouse mAb	1:100	Santa Cruz
	CD79a	HM57	mouse mAb	1:75	Exbio
	CD79a	HM57	mouse mAb	1:75	Novusbio
	CD20		rabbit pAb	1:100	ThermoFisher
T lymphocytes	CD3		rabbit pAb	1:200	Dako
	CD3	F7.2.38	mouse mAb	1:50	Santa Cruz
T-reg lymphocytes	Foxp3	eBio7979	mouse mAb	1:50	eBioscience
Macrophages	Iba-1		rabbit pAb	1:300	Wako
	Calprotectin	MAC387	mouse mAb	1:50	ThermoFisher
	CD163	AM-3K	mouse mAb	1:50	Transgenic Inc
	CD68	PG-M1	mouse mAb	1:50	ThermoFisher

**Table 2 animals-12-03064-t002:** Summary of the immunopositive cell distribution in different areas of the secondary lymphoid organs of the roe and fallow deer analyzed. Results expressed on the percentage of immunostained cells (-, absence; +, <10%; ++, 10–50%; +++, 50–90%; ++++, >90%).

Antibody		RLN ^1^					Spleen			
		MD ^2^	PC ^3^	GC ^4^	MTL ^5^	IFZ ^6^	Red Pulp	White Pulp		
								PALS ^7^	GC	MTL
Roe deer										
CD79a	Exbio	-	-	++	+++	-	-	-	++	+++
CD79a	Novusbio	-	-	+	++	-	-	-	+	++
CD20	TermoFisher	-	+	+++	++++	-	-	-	+++	++++
CD3	Dako	-	+++	+	-	++++	-	+++	+	-
CD3	Santa Cruz	-	++	+	-	++++	-	++	+	-
Iba-1	Wako	+++	++	+	-	++	++++	+	+	-
MAC387	ThermoFisher	++	+	-	-	+	++	+	-	-
CD163	ThermoFisher	++	+	-	-	-	++	+	-	-
Fallow deer										
CD79a	Exbio	-	-	++	++++	-	-	-	++	++++
CD79a	Novusbio	-	-	+	++	-	-	-	+	++
CD20	TermoFisher	-	+	++++	++++	-	-	-	++++	++++
CD3	Dako	-	+++	+	-	++++	-	+++	+	-
CD3	Santa Cruz	-	++	+	-	+++	-	++	+	-
Iba-1	Wako	+++	++	+	-	++	++++	+	+	-
MAC387	ThermoFisher	++	+	-	-	+	++	+	-	-
CD163	ThermoFisher	+++	+	-	-	-	+++	+	-	-

^1^ Retromandibular Lymph Nodes; ^2^ Medulla; ^3^ Paracortex; ^4^ Germinal Center; ^5^ Mantle Zone; ^6^ Interfollicular Zone; ^7^ Periarteriolar Lymphoid Sheath.

## Data Availability

The data presented in this study are available on request from the corresponding author.

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
