# Peer review of "Immunohistochemical Characterization of Immune System Cells in Lymphoid Organs from Roe and Fallow Deer"

_animals, 2022, doi:10.3390/ani12213064_

Round 1
Reviewer 1 Report
I have no minor nor major comments on this paper, it is well written and presented with sound conclusions. The only thig I may miss are controls of the antibodies staining human tissue, and whether or not the antibodies might stain frozen material, but I do not know whether frozen samples an be obtained.
Author Response
We appreciate the reviewer’s insightful suggestion and agree that frozen material represent surely a better specimen concerning immunoreactivity for many antigens. However, since all of these antibodies were commercial antibodies widely used in FFPE canine samples as reported in their respective datasheet and in many reports available in the literature [24,25], we deemed that such positive controls could be suitable for the study’s aim.
Reviewer 2 Report
Comments:
Section 1 Introduction
More emphasis in the Introduction about the actual contribution of this study to the field is require.
Section 3 Results
1. Why are there no pictures of H&E staining in the manuscript. H&E histological staining should be shown together with immunohistochemical (IHC) staining?
2. The images of IHC (Figure 1; Figure 2; Figure 3) should additionally include images taken at higher magnification.
3. Why in the IHC pictures (Figure 1; Figure 2; Figure 3) the black line segment corresponding to bar=50µm; bar=100µm; bar=200µm is the same length? Please explain in detail how this is possible!
4. For all tested antibodies, a Western blot analysis should additionally be performed which would significantly increase the value of the scientific study.
5. The sentence "No differences in immunoreactivity between the roe and fallow deer tissues were highlighted" is unfounded. In my opinion, statistical analysis should be performed.
6. The results of the immunopositive cell distribution in different areas of the secondary lymphoid organs in cervids (Table 2) should be shown separately for roe deer and fallow deer. Data from two different species cannot be combined. This is incorrect!
Section 4 Discussion
The discussion contains too little publication data on the specificity of these studies.
Author Response
Section 1 Introduction
- More emphasis in the Introduction about the actual contribution of this study to the field is require.
As suggested by the reviewer we have stressed more this issue both in the introduction (lines 58-70) and also in the discussion section (lines 264-276). Additionally, very recent bibliographic references that emphasize the value of a multimodal diagnostic strategy for wild ungulates are added [4,12,15,16,35,36].
Section 3 Results
- Why are there no pictures of H&E staining in the manuscript. H&E histological staining should be shown together with immunohistochemical (IHC) staining?
H&E stained histological section pictures have been added in each of the three figures, as suggested.
- The images of IHC (Figure 1; Figure 2; Figure 3) should additionally include images taken at higher magnification.
High-magnification inserts have been added to the IHC pictures in each figure.
- Why in the IHC pictures (Figure 1; Figure 2; Figure 3) the black line segment corresponding to bar=50µm; bar=100µm; bar=200µm is the same length? Please explain in detail how this is possible!
We thank the reviewer for pointing out this typo. The bar length in all the figures have been modified.
- For all tested antibodies, a Western blot analysis should additionally be performed which would significantly increase the value of the scientific study.
We appreciate the reviewer’s insightful suggestion and agree that WB could be useful to further detect the epitopes investigated; however, we believe that, according to many other analogs studies [14,19,26,30,32], the lymphoid region distribution inside the organ architecture, the stained cell morphology, and the IHC staining patterns of the tested antibodies (cytoplasmic, membranous, nuclear) adequately support our arguments. Nevertheless, we recognize this limitation should be mentioned in the paper, so we added the following sentence in lines “Additional analytical methods, such as western blot (WB) or protein sequencing by mass spectrometry, could be used to demonstrate the detection of identical epitopes; nonetheless, lymphoid region distribution, cell morphology, and staining patterns of the tested antibodies allow us to confirm the specificity of our findings, as reported elsewhere [14,19,26,30].”
- The sentence "No differences in immunoreactivity between the roe and fallow deer tissues were highlighted" is unfounded. In my opinion, statistical analysis should be performed.
We thank the reviewer for pointing this out. As suggested, the sentence has been deleted. Since the histological description in the results section is valid in both species and there are only very slight differences in the yield of antibodies, we have added a sentence to lines 177-179 " …cervine RLN and spleen to varying degrees of intensity and the yield was similar in both species" and split the table 2 showing the semiquantitative score divided by roe and fallow deer to present the results more clearly.
- The results of the immunopositive cell distribution in different areas of the secondary lymphoid organs in cervids (Table 2) should be shown separately for roe deer and fallow deer. Data from two different species cannot be combined. This is incorrect!
As suggested by the reviewer, we have rearranged Table 2, which includes the results of the immunopositive cell distribution in different areas of the lymphoid tissue for each marker, showing the data separately for roe and fallow deer.
Section 4 Discussion
- The discussion contains too little publication data on the specificity of these studies.
Thanks for pointing this out. Unfortunately, we acknowledge the limited amount of bibliographic data concerning the immunophenotyping of lymphoid cells in these species. The abstract, introduction, and discussion sections draw attention to this knowledge gap. Moreover, as suggested by the reviewer, we added a new paragraph (265-271) with more bibliographic references on studies carried out in other species of metatherians and eutherians with an analogous methodology [14,26,31-34]. The study's rationale (lines 268-271), the framework concerning the health monitoring of roe and fallow deer (lines 275-277), and how this study could provide useful information are all further clarified. In addition, very recent bibliographic references on this issue are added throughout the text.
Reviewer 3 Report
A very interesting article covering two major European wild ruminant species. The applied histochemical methods do not raise my doubts.
The layout of the article and the way of description is typical for scientific and research works.
As a hunting practice and veterinarian, great doubts arise from the assessment of health condition only on the basis of macroscopic examination.
Very rarely, free-living individuals are free from internal parasites - the only question is how severe the invasion is. Additionally, no information is available at what time of year the males and females were obtained?
In my opinion, the so-called null sample, e.g. preparations of diseased lymph nodes and spleen.
Therefore, I believe that the statement in the conclusions that it is a non-pathological secondary lymphoid organs tissue - is definitely exaggerated.
Author Response
We thank the reviewer for the time spent to review the manuscript. We appreciate your valuable comments and suggestion.
More details about the time of the year in which the animals were shot have been added to the text in line 80 “…samples were collected during the 2020-2021 hunting season, in the months of February and March, from…”
In regard to the assessment of health conditions, we agree that the absence of macroscopic lesions does not rule out an absence of histopathological changes. However, an evaluation of the general health status of the animals was not the aim of the study, so we modified the conclusive statement in line 360, and, as suggested by the reviewer, we have deleted the term “non-pathological” throughout all the text and added the milder sentence “without recognizable pathological changes”. Moreover in lines (87-89) we have added the sentence “The samples were also examined histologically, and those that showed obvious pathological changes were excluded.” to present the methodology more clearly.
Round 2
Reviewer 2 Report
Comments:
Section 3 Results:
1. On the histological H&E staining images, the authors should indicate the visible structures and also the key cells that are identified/diagnosed. In addition, H&E staining images at higher magnification should be added.
2. What is the magnification or bar scale of the added IHC staining images. Please write.
3. In the description of Figure 1, Figure 2 and Figure 3 the bar scale=100 µm is given for all images and the length of the black section in the images has different lengths. This is a serious methodological error. How is this possible. Previously, in the description of Figure 1, Figure 2 and Figure 3 it wrote: bar=50µm; bar=100µm; bar=200µm and the black sections in the pictures were the same length. I have doubts about the reliability of these studies.
Author Response
- On the histological H&E staining images, the authors should indicate the visible structures and also the key cells that are identified/diagnosed. In addition, H&E staining images at higher magnification should be added.
We thank the Reviewer for the suggestion. In the figures arrows and arrow heads have been added to H&E staining images to highlight the visible structures as suggested. We believe that the addition of inserts would not give further information about the lymph node architecture, as the different cells are not distinguishable in a section stained with H&E.
- What is the magnification or bar scale of the added IHC staining images. Please write.
In the insert of IHC staining images bars showing the magnification have been added.
- In the description of Figure 1, Figure 2 and Figure 3 the bar scale=100 µm is given for all images and the length of the black section in the images has different lengths. This is a serious methodological error. How is this possible. Previously, in the description of Figure 1, Figure 2 and Figure 3 it wrote: bar=50µm; bar=100µm; bar=200µm and the black sections in the pictures were the same length. I have doubts about the reliability of these studies.
In the figures 1, 2 and 3 the images were taken with different objectives. The images 1A, 1B, 1C, 1F, 2A, 2B, 3A and 3B were taken with a 20x objective, while the other images were taken with a 40x objective for this reason in the figures the 100 mm bar have a different length. In the previous revision we have unified the length of the bars in the figure captions and obviously changed the bar length in the images. For the bars put in the inserts we have carefully checked the magnifications.